# SCALABLE HIERARCHICAL EMBEDDINGS OF COMPLEX NETWORKS

## ABSTRACT

Graph representation learning has become important in order to understand and predict intrinsic structures in complex networks. A variety of embedding methods has in recent years been developed including the Latent Distance Modeling (LDM) approach. A major challenge is scaling network embedding approaches to very large networks and a drawback of LDM is the computational cost invoked evaluating the full likelihood having $\mathcal{O}(N^2)$ complexity, making such analysis of large networks infeasible. We propose a novel multiscale hierarchical estimate of the full likelihood of LDMs providing high-details where the likelihood approximation is most important while scaling in complexity at $\mathcal{O}(N \log N)$. The approach relies on a clustering procedure approximating the Euclidean norm of every node pair according to the multiscale hierarchical structure imposed. We demonstrate the accuracy of our approximation and for the first time embed very large networks in the order of a million nodes using LDM and contrast the predictive performance to prominent scalable graph embedding approaches. We find that our approach significantly outperforms these existing scalable approaches in the ability to perform link prediction, node clustering and classification utilizing a surprisingly low embedding dimensionality of two to three dimensions whereas the extracted hierarchical structure facilitates network visualization and interpretation. The developed scalable hierarchical embedding approach enables accurate low dimensional representations of very large networks providing detailed visualizations that can further our understanding of their properties and structure.

## 1 INTRODUCTION

Networks naturally arise in a plethora of scientific areas to model the interactions between entities from physics to sociology and biology, with many instances such as collaboration, protein-protein interaction, and brain connectivity networks (Newman, 2003). In recent years Graph Representation Learning (GRL) approaches have attracted great interest with their outstanding performance compared to the classical techniques for the challenging network analysis problems such as link prediction (Liben-Nowell & Kleinberg, 2003; Backstrom & Leskovec, 2011), node classification (Getoor & Taskar, 2007; Grover & Leskovec, 2016), and community detection (Fortunato, 2010).

Many existing GRL methods (Hamilton et al., 2017b; Zhang et al., 2020) mainly aim to capture the underlying intrinsic relationships among the nodes by either performing random walks (Perozzi et al., 2014; Grover & Leskovec, 2016) over the network or designing a matrix capturing the first and high order node proximities (Cao et al., 2015; Ou et al., 2016). However, they require high computational and space costs because of the exact node sampling procedures or the expensive factorization of dense proximity matrices. The recent Graph Neural Networks (GNNs) (Hamilton et al., 2017b; Zhang et al., 2020; Wang et al., 2016) methods provide effective tools in learning the node representations by leveraging the side information such as node attribute features; nevertheless, they also face computational difficulties, especially for large-scale networks consisting of millions of nodes and edges. Although the recent studies aim to alleviate the computational burden of the algorithms through matrix sparsification tools (Qiu et al., 2019) or hierarchical representations (Bhowmick et al., 2020; Chen et al., 2018), the performance of the methods in the downstream tasks significantly drops, and they require larger embedding sizes to compensate for the loss.

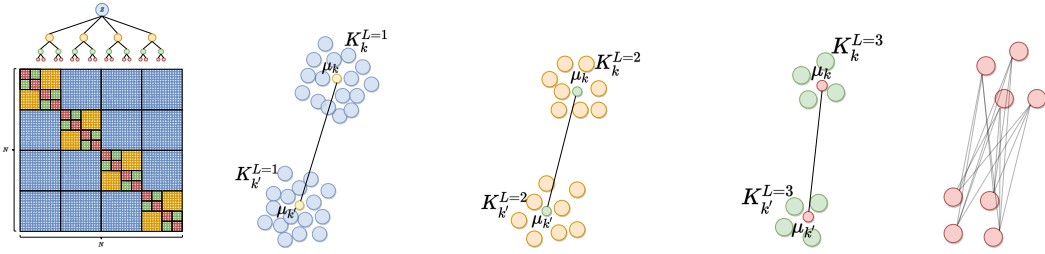

Figure 1: Schematics of the distance matrix calculation for a hierarchical structure of tree height $L = 3$ and number of observations $N = 64$.

Latent Space Models (LSMs) for the representation of graphs have been quite popular over the past years, especially for social networks analysis (Hoff et al., 2002). LSMs utilize the generalized linear model framework to obtain informative latent node embeddings while preserving network characteristics. The choice of latent effects in modeling the link probabilities between the nodes leads to different expressive capabilities characterizing network structure. We consider the Latent Distance Model (LDM) (Hoff et al., 2002) with the Euclidean norm, in which nodes are placed closer in the latent space if they are similar or vice-versa. LDM obeys the triangle inequality and thus naturally represents transitivity and network homophily. These methods are attractive due to their simplicity, as they define well-structured inference problems and are characterized by high explanatory power. The time and space complexities are their main drawbacks, which scale quadratically with the number of nodes in the graph.

Many real-world networks can be expressed as hierarchical structures of different scales (Ravasz & Barabási, 2003). For this purpose, several hierarchical network modeling tools have been proposed, such as the extensions of the stochastic block model to binary and multifurcating hierarchical structures (Clauset et al., 2008; Roy et al., 2007; Blundell et al., 2012; Herlau et al., 2012; 2013) as well as agglomerative (Blondel et al., 2008; Ahn et al., 2010) and recursive partitioning procedures (Li et al., 2020) relying on various measures of similarity. Learning the node representations preserving the hierarchical structure of the network is also a very promising task, and it can facilitate the visualization and the understanding of the inner dynamics of the network.

In this work, we propose the Scalable Hierarchical Latent Distance Model (SH-LDM) combining embedding and hierarchical representations for graph representation learning. Importantly, the hierarchical structure imposed in (SH-LDM) reduces the total time and space complexity of the LDM to linearithmic in terms of the number of nodes (i.e., $\mathcal{O}(N \log N)$) at the same time providing accurate interpretable representation of structure at different scales. Using the SH-LDM we embed moderate sized and large-scale networks containing more than a million nodes and establish the performance of LDM in terms of link prediction and node classification to existing prominent scalable graph embedding approaches. We further highlight how the inferred hierarchical organization can facilitate accurate visualization of network structure even when using only $D = 2$ dimensional representations providing favorable performance in all the considered GRL tasks; link-prediction, node classification, node clustering, and network reconstruction.

In summary, our contributions are to reconcile embedding and hierarchical representations providing accurate linearithmic approximation of the full likelihood, efficient inference, enhanced visualization and network compression utilizing ultra-low embedding dimensions and hierarchical representations.

## 2 THE SCALABLE HIERARCHICAL-LATENT DISTANCE MODEL

We presently concentrate our study on the case of undirected networks, but we note that our approach generalizes to both directed and bipartite graphs as described in the supplementary material. Let $\mathcal{G} = (V, E)$ be a graph where $N := |V|$ is the number of nodes and $Y_{N \times N} = [y_{i,j}]$ be the adjacency matrix of the graph such that $y_{i,j} = 1$ if there is an edge between the nodes $v_i$ and $v_j$ and otherwise it is equal 0 for all $1 \leq i < j \leq N$.

A Latent Space Model (LSM) defines a $\mathcal{R}^D$-dimensional latent space in which every node of the graph is characterized through the unobserved but informative node-specific variables $\{\mathbf{z}_i \in \mathcal{R}^D\}$. These variables are considered sufficient to describe and explain the underlying relationships between the nodes of the network, such as transitivity and homophily. The probability of an occurring edge between an ordered pair of the graph is considered conditionally independent given the unobserved latent positions. Consequently, the total probability distribution of the network can be written as:

$$P(Y|\boldsymbol{Z}, \boldsymbol{\theta}) = \prod_{i<j}^{N} p(y_{i,j}|\mathbf{z}_i, \mathbf{z}_j), \tag{1}$$

A popular and convenient parameterization of equation 1 for binary data is through the logistic regression model (Hoff et al., 2002; Handcock et al., 2007; Krivitsky et al., 2009; Hoff, 2005). In contrast, we adopt the Poisson regression model as proposed in Hoff (2005) under a generalized linear model framework for the LSM. The use of a Poisson likelihood for modelling binary relationships in a network does not decrease the predictive performance nor the ability of the model to detect the network structure, as shown in Wind & Mørup (2012) and also generalize the analysis to integer weighted graphs. In addition, the exchange of the *logit* to a *log* link function when transitioning from a Bernoulli to a Poisson model defines nice decoupling properties over the predictor variables in the likelihood (Karrer & Newman, 2011; Herlau et al., 2014). Utilizing the Poisson Latent Distance Model (LDM) of the LSM family framework, the rate of an occurring edge depends on a distance metric between the latent positions of the two nodes. We consider the LDM with node-specific biases or random-effects (Hoff, 2005; Krivitsky et al., 2009) such that the expression for the Poisson rate becomes:

$$\lambda_{ij} = \exp\left(\gamma_i + \gamma_j - d\big(\mathbf{z}_i, \mathbf{z}_j\big)\right). \tag{2}$$

where $\gamma_i$ denotes the node-specific random-effects and $d_{ij}(\cdot, \cdot)$ denotes any distance metric obeying the triangle inequality $\left\{ d_{ij} \leq d_{ik} + d_{kj},\ \forall (i, j, k) \right\}$. Considering variables $\mathbf{z}$ as the latent characteristics, Eq. equation 2 shows that similar nodes will be placed closer in the latent space, yielding a high probability of an occurring edge and thus modeling homophily and satisfies network transitivity and reciprocity through the triangle inequality whereas the node specific bias can account for degree heterogeneity. The conventional LDM utilizing a global bias, $\gamma^g$, corresponds to the special case in which $\gamma_i = \gamma_j = 0.5\gamma^g$. As in Hoff et al. (2002), we presently adopt the Euclidean distance as the choice for the distance metric $d_{ij}(\cdot, \cdot)$.

## 2.1 SCALING THE LATENT DISTANCE MODEL

Optimizing the LDM requires the computation of the log-likelihood which is defined as the sum over each ordered pair of the network as:

$$\log P(Y|\boldsymbol{\lambda}) = \sum_{i<j} \left( y_{ij} \log(\lambda_{ij}) - \lambda_{ij} \right) = \sum_{i<j:y_{ij}=1} \log(\lambda_{ij}) - \sum_{i<j} \lambda_{ij}, \tag{3}$$

For brevity, we presently ignore the linear scaling of the above log-likelihood by dimensionality $D$. Large networks are highly sparse (Barabási & Pósfai, 2016) with the number of edges for very sparse networks being proportional to the number of nodes in the network. As a result, the computation of the link contribution $\sum_{y_{i,j}=1} \log(\lambda_{i,j})$ is relatively cheap scaling linearithmic or sub-linearithmic (see also supplementary material). This is not the case for the second term which still requires the computation of all node pairs scaling as $\mathcal{O}(N^2)$ making the evaluation of the above likelihood infeasible for large networks.

To reduce the complexity, we propose to approximate the non-link term $\mathcal{O}(N^2)$ using blocks, i.e., akin to stochastic block models White et al. (1976); Holland et al. (1983); Nowicki & Snijders (2001), in which we when grouping the nodes into $K$ clusters define the rate between block $k$ and $k'$ in terms of their distance between centroids,

$$\sum_{i<j} \lambda_{ij} \approx \sum_{k}^{K} \left( \sum_{i,j \in C_k} e^{(\gamma_i + \gamma_j - ||\mathbf{z}_i - \mathbf{z}_j||_2)} + \sum_{i \in k, j \notin C_k} e^{(\gamma_i + \gamma_j - ||\boldsymbol{\mu}_k - \boldsymbol{\mu}_{k'}||_2)} \right)$$

$$= \left( \sum_{k}^{K} \sum_{i,j \in C_k} e^{(\gamma_i + \gamma_j - ||\mathbf{z}_i - \mathbf{z}_j||_2)} \right) + \sum_{k > k'} e^{-||\boldsymbol{\mu}_k - \boldsymbol{\mu}_{k'}||_2} \left( \sum_{i \in C_k} e^{\gamma_i} \right) \left( \sum_{j \in C_{k'}} e^{\gamma_j} \right), \quad (4)$$

where $\mu_k$ denotes the $k$'th cluster centroid of the set $\boldsymbol{C} = \{C_1, \ldots, C_K\}$, and has absorbed the dependency over the variables $\boldsymbol{Z}$.

Overall, considering the main principle of the LDM that connected and homophilic nodes will be placed closer in the latent space, this approximation generalizes this principle introducing a clustering procedure that obeys "cluster-homophily" and "cluster-transitivity" over the latent space. More specifically, we can assume that closely related nodes will be positioned in the same cluster while related or interconnected clusters will also be positioned close in the latent space, providing an accurate approximation schema. Assuming equally sized clusters having $N/K$ nodes the first part scales $\mathcal{O}(N^2/K)$ whereas the second part scales $\mathcal{O}(K^2)$. As such, there is an undesirable inherent trade-off in which the first term reduces by $K$ but the second term increases quadratically. Thus, by setting $K = N/log(N)$ we reduce the first part to scale as $\mathcal{O}(N \log N)$ but at the cost of the second term scaling $\mathcal{O}(N^2/log(N)^2)$ which for large networks is still prohibitive.

### 2.1.1 HIERARCHICAL APPROXIMATION OF THE LATENT DISTANCE MODEL

For the second term to be scalable, we need for $K = N/log(N)$ the contribution for the second term to scale also at $\mathcal{O}(N \log N)$. To achieve this, we adopt a multiresolution KD tree (mrkd-tree) structure similar to Moore (1999) which endorses a hierarchical divisive clustering architecture. The root of the tree contains the total amount of latent variables $\boldsymbol{Z}$. At every level of the tree, we perform partitioning of the tree-nodes which are not considered as leafs. The tree-nodes belonging to a specific level are considered as the clusters for that specific tree height. Every novel partition of a non-leaf node is performed only on the set of points allocated to the parent tree-node (cluster). A node is considered as a leaf if the corresponding cluster contains less than a specific amount of datapoints which for $K = N/log(N)$ is set to be approximately equal to $N/K = \log N$. For every level of the tree, we consider the pairwise distances of datapoints belonging to different tree-nodes as the distance between the corresponding cluster centroids, as illustrated by Figures 1b, 1c and 1d. Based on these distances, we calculate the likelihood contribution of these approximation blocks and continue down the tree for the non-leaf tree-nodes. In the last level (or when all tree-nodes are considered as leafs) we calculate analytically the inner cluster pairwise distances for the corresponding likelihood contribution of analytical blocks, as shown by Figure 1e.

We can thereby define a Scalable Hierarchical-Latent Distance Model with Random Effects (SH-LDM-RE) as:

$$\log P(Y|\boldsymbol{\lambda}) \quad = \sum_{y_{i,j}=1} \left( \gamma_i + \gamma_j - ||\mathbf{z}_i - \mathbf{z}_j||_2 \right) - \sum_{k_L}^{K_L} \left( \sum_{i,j \in C_{k_L}} e^{(\gamma_i + \gamma_j - ||\mathbf{z}_i - \mathbf{z}_j||_2)} \right) \quad (5)$$

$$- \sum_{l}^{L} \left( \sum_{k_l > k_l'} e^{-||\boldsymbol{\mu}_{k_l} - \boldsymbol{\mu}_{k_l'}||_2} \left( \sum_{i \in C_{k_l}} e^{\gamma_i} \right) \left( \sum_{j \in C_{k_l'}} e^{\gamma_j} \right) \right), \quad (6)$$

where $l = [1, \cdots L]$ denotes the $l$'th mrkd-tree level, $k_l$ is the index representing the cluster id for the different tree levels and $\boldsymbol{\mu}_{k_l}$ the corresponding centroid. We also consider a Scalable Hierarchical-Latent Distance Model (SH-LDM) without the random effects setting $\gamma_i = 0.5\gamma^g$. For a multi-furcating tree splitting in $C$ clusters and having $N/log(N)$ terminal nodes (clusters) the number of internal nodes are $\mathcal{O}(N/(C \log N))$ and each node needs to evaluate $\mathcal{O}(C^2)$ pairs providing an overall complexity of $\mathcal{O}(NC/\log N)$, thus $C \leq \log N^2$ to achieve $\mathcal{O}(N \log N)$ scaling (Epp, 2010).

### 2.1.2 DIVISIVE PARTITIONING USING K-MEANS WITH A EUCLIDEAN DISTANCE METRIC

Whereas the likelihood in equation 6 can be directly minimized by assigning nodes to the clusters given by the tree structure this evaluation for all $N$ nodes scales prohibitively as $\mathcal{O}(N^2/\log N)$.

To reduce this scaling we use a more efficient divisive partitioning procedure. For efficiency instead of considering minimizing the full expression in equation 6 we consider minimizing the Euclidean norm $||\boldsymbol{\mu}_{k_l} - \boldsymbol{\mu}_{k'_l}||_2$. The divisive clustering procedure thus relies on the following Euclidean norm objective

$$J(\mathbf{r}, \boldsymbol{\mu}) = \sum_{i=1}^{N} \sum_{k=1}^{K} r_{ik}||\mathbf{z}_i - \boldsymbol{\mu}_k||_2, \tag{7}$$

where $k$ denotes the cluster id, $\mathbf{z}_i$ is the i'th data observation, $r_{ik}$ the cluster responsibility/assignment, and $\boldsymbol{\mu}_k$ the cluster centroid.

This objective function is unfortunately not accounted for by existing k-means clustering algorithms relying on the squared Euclidean norm. We therefore presently derive an optimization procedure for k-means clustering with Euclidean norm utilizing the auxiliary function framework of Tsutsu & Morikawa (2012) developed in the context of compressed sensing. We define an auxiliary function for Eq. equation 7 as:

$$J^{+}(\boldsymbol{\phi}, \mathbf{r}, \boldsymbol{\mu}) = \sum_{i=1}^{N} \sum_{k=1}^{K} r_{ik} \Big( \frac{||\mathbf{z}_i - \boldsymbol{\mu}_k||_2^2}{2\phi_{ik}} + \frac{1}{2}\phi_{ik} \Big), \tag{8}$$

where $\phi$ are the auxiliary variables. Thereby, minimizing Eq. equation 8 with respect to $\phi_{nk}$ yields $\phi_{ik}^* = ||\mathbf{z}_i - \boldsymbol{\mu}_k||_2$ and by plugging $\phi_{ik}^*$ back to Eq. equation 7 we obtain $J^{+}(\boldsymbol{\phi}^*, \mathbf{r}, \boldsymbol{\mu}) = J(\mathbf{r}, \boldsymbol{\mu})$ verifying that Eq. equation 8 is indeed a valid auxiliary function for Eq. equation 7. The algorithm proceeds by optimizing cluster centroids as $\boldsymbol{\mu}_k = \Big( \sum_{i \in k} \frac{\mathbf{x_i}}{\phi_{ik}} / \sum_{i \in k} \frac{1}{\phi_{ik}} \Big)$ and assigning points to centroids as $\arg\min_{\boldsymbol{C}} = \sum_{k=1}^{K} \sum_{\mathbf{z} \in C_k} \Big( \frac{||\mathbf{z} - \boldsymbol{\mu}_k||_2^2}{2\phi_k} + \frac{1}{2}\phi_k \Big)$ upon which $\phi_k$ is updated. The overall complexity of this procedure is $\mathcal{O}(TKND)$ (Hartigan & Wong, 1979) where $T$ is the number of iterations required to converge.

A simple approach to construct the tree structure would be to use the above Euclidean k-means procedure to split the nodes into $K = N/\log(N)$ clusters and construct the tree according to agglomeration as in hierarchical clustering. Unfortunately, such a strategy is prohibitive for $K = N/\log N$, and maximally $K = \log N$ clusters can be obtained for scaling of $\mathcal{O}(N \log N)$. It would be tempting to continue splitting into $\log N$ clusters, however, for a balanced multifurcating tree with $N/\log N$ leaf clusters this will result in a height scaling as $\mathcal{O}(\log N/\log\log N)$ and thus an overall complexity of $\mathcal{O}(N \log N^2/\log\log N)$ (Epp, 2010). Whereas using a balanced binary tree at all levels below the root results in a height scaling as $\mathcal{O}(\log N)$ providing an overall complexity when including the linear scaling by dimensionality $D$ of $\mathcal{O}(DN \log N)$ for the SH-LDM. Figure 1a illustrates the resulting tree for a small problem of $N = 64$ nodes in which we first split into 4 ($\approx \log(64)$) clusters and subsequently create binary splits until each cluster contains 4 ($\approx \log(64)$) nodes.

## 2.2 RELATED WORK

In recent years, we have witnessed a tremendous increase in the number of GRL methods. The leading initial works are the random walk-based methods (Perozzi et al., 2014; Grover & Leskovec, 2016; Tang et al., 2015; Nguyen & Malliaros, 2018), which generate fixed-length node sequences by following a strategy and leverages the Skip-Gram algorithm (Mikolov et al., 2013) to learn the node representations. Another prominent class of techniques is the matrix factorization-based algorithms (Cao et al., 2015; Ou et al., 2016; Zhang et al., 2020). They extract the embedding vectors by decomposing a designed feature matrix. The neural networks are undoubtedly one of the most outstanding models, and many models (Hamilton et al., 2017b; Wang et al., 2016; Zhang et al., 2020; Cao et al., 2016; Vincent et al., 2010; Kipf & Welling, 2017; Hamilton et al., 2017a) allowing to incorporate the node attributes with the network structure when learning the embeddings have been proposed for graph-structured data. Although these approaches are quite effective in downstream tasks, they require a high computational burden, making them inapplicable for large-scale networks. Therefore, the recent works (Bhowmick et al., 2020; Zhang et al., 2019) capitalize on the computational challenges emerging for large networks by either incorporating fast approximation techniques with the existing models (Qiu et al., 2019) or developing novel approaches relying on fast hashing schemes (Yang et al., 2019; Zhang et al., 2018). Our approach is related to current efforts in scaling graph embedding approaches for the analysis of very large networks.

Table 1: Statistics of networks. $N$: number of nodes, $M$: number of edges.

|   | Cora | Dblp | AstroPh | GrQc | Facebook | HepTh | Amazon | YouTube | Flickr | Flixster |
|---|------|------|---------|------|----------|-------|--------|---------|--------|----------|
| $N$ | 2,708 | 27,199 | 17,903 | 5,242 | 4,039 | 8,638 | 334,868 | 1,138,499 | 1,715,255 | 2,523,386 |
| $M$ | 5,278 | 66,832 | 197,031 | 14,496 | 88,234 | 24,827 | 925,876 | 2,990,443 | 15,555,042 | 7,918,801 |

Table 2: Area Under Curve scores for representation sizes of 2 and 8 over moderate-sized networks.

|  | AstroPh | | GrQc | | Facebook | | HepTh | | Cora | | DBLP | |
|---|------|------|------|------|------|------|------|------|------|------|------|------|
| Dimension ($D$) | 2 | 8 | 2 | 8 | 2 | 8 | 2 | 8 | 2 | 8 | 2 | 8 |
| DEEPWALK | .831 | .945 | .845 | .919 | .958 | .986 | .773 | .874 | .684 | .782 | .803 | .939 |
| NODE2VEC | .825 | .950 | .809 | .884 | .914 | **.988** | .780 | .881 | .640 | .776 | .803 | .945 |
| LINE | .632 | .910 | .688 | .920 | .751 | .980 | .659 | .874 | .634 | .521 | .625 | .503 |
| NETMF | .800 | .814 | .830 | .860 | .872 | .935 | .757 | .792 | .629 | .739 | .838 | .858 |
| NETSMF | .828 | .891 | .756 | .805 | .907 | .976 | .705 | .810 | .605 | .737 | .766 | .857 |
| RANDNE | .524 | .554 | .534 | .560 | .614 | .657 | .519 | .509 | .508 | .556 | .508 | .517 |
| LOUVAINNE | .798 | .813 | .861 | .868 | .957 | .958 | .774 | .874 | .767 | .747 | .900 | .904 |
| PRONE | .768 | .907 | .818 | .883 | .900 | .971 | .678 | .823 | .675 | .764 | .813 | .924 |
| SH-LDM | .917 | .960 | .898 | .944 | .980 | .986 | .847 | .912 | **.786** | .792 | .919 | .956 |
| SH-LDM-RE | **.931** | **.962** | **.910** | **.953** | **.986** | **.988** | **.869** | **.923** | .759 | **.795** | **.930** | **.963** |

The most related scalable study to our method is the work for the LDM Raftery et al. (2012) where the $\mathcal{O}(N^2)$ complexity is lowered to $\mathcal{O}(cE)$ with $c \ll N$ via a case-control approach motivated by epidemiology studies. While the overall complexity of the case-control model can scale the analysis of LDMs to massive networks creating the All-Pairs Shortest Paths matrix that is required to perform the proposed stratified sampling over non-links, is not possible for large networks, both in terms of time and space complexity ($> \mathcal{O}(N^2)$) and the authors did not consider networks larger than $N = 2716$ nodes. In the supplementary material, we therefore examine a scalable version of the case-control approach. In order to scale the case-control model to the order of millions of nodes as presently considered we exchanged the stratified sampling with a uniform sampling procedure which as argued by the authors may not be the best estimator. We conclude that also the case-control approach can be used to successfully scale the LDMs. Notably, the case-control approach does not yield a hierarchical representation of network structure as the presently developed SH-LDM. However, the developed hierarchical clustering procedure can be used as a post-processing step to extract hierarchical representations from the learned embeddings.

## 3 EXPERIMENTS

We extensively evaluate the performance of our approach compared to other baseline approaches on ten networks of various sizes and structures. We consider each network as undirected and unweighted for the consistency of the experiments, and the detailed statistics of the networks are given in Table 1. We have conducted all the experiments regarding the SH-LDM and SH-LDM-RE on a 32 GB Tesla V100 GPU machine. For the SH-LDM and SH-LDM-RE models, we optimize the negative log-likelihood via the Adam (Kingma & Ba, 2017) optimizer while setting the learning rate to $lr = 0.1$ for all networks. The parameters of the baseline approaches have been tuned. Because of the limitation in the number of pages, the details of the parameter settings, the datasets as well as a scalable spectral clustering initialization scheme for SH-LDM are provided as supplementary materials. In the following we report results on the prominent graph representation learning tasks; link prediction, node classification, and network visualization. For the network visualization we further include quantitative evaluations in terms of network reconstruction and node clustering quality.

**Link Prediction:** We report results for the area under the curve of the receiver operator characteristic (AUC) whereas we defer the corresponding Precision-Recall AUC to the supplementary material. For the experimental setup, we follow the commonly applied strategy (Perozzi et al., 2014; Grover & Leskovec, 2016), and we remove half of the edges of a given network by keeping the residual network connected. For the large-scale networks , this approach is infeasible, so we apply a scalable evaluation technique (Bhowmick et al., 2020; Zhang et al., 2018). We hide 30% of the edges of a

Table 3: Area Under Curve scores for varying representation sizes over the large-scale networks. The symbol "-" indicates that the running time of the corresponding model takes more than 20 hours and "x" shows that the method is not able to run due to insufficient memory space.

| Dimension ($D$) | *Amazon* | | | *YouTube* | | | *Flickr* | | | *Flixster* | | |
|---|---|---|---|---|---|---|---|---|---|---|---|---|
| | 2 | 3 | 8 | 2 | 3 | 8 | 2 | 3 | 8 | 2 | 3 | 8 |
| DEEPWALK | .839 | .932 | .972 | .822 | .891 | .921 | .889 | .937 | .972 | .820 | .866 | .921 |
| NODE2VEC | .813 | .880 | .968 | - | - | - | - | - | - | - | - | - |
| LINE | .626 | .501 | .500 | .660 | .832 | .878 | .685 | .889 | .921 | .523 | .868 | .936 |
| NETMF | .829 | .831 | .858 | x | x | x | x | x | x | x | x | x |
| NETSMF | .768 | .786 | .835 | .939 | .940 | .949 | .974 | .977 | .980 | **.987** | **.987** | **.987** |
| RANDNE | .507 | .511 | .514 | .672 | .700 | .762 | .833 | .869 | .903 | .700 | .739 | .835 |
| LOUVAINNE | .955 | .954 | .954 | .820 | .819 | .815 | .898 | .899 | .909 | .735 | .718 | .746 |
| PRONE | .847 | .901 | .944 | .691 | .761 | .861 | .623 | .819 | .908 | .756 | .803 | .846 |
| SH-LDM | .974 | .980 | .988 | .899 | .920 | .935 | .972 | .979 | .986 | .897 | .916 | .932 |
| SH-LDM-RE | **.976** | **.981** | **.988** | **.940** | **.947** | **.957** | **.980** | **.985** | **.988** | .962 | .969 | .971 |

given initial network and consider the greatest connected component (GCC) of the residual network to learn the node representations. We utilize the residual network to learn the node embeddings and report the best performing binary operator (Grover & Leskovec, 2016) used for logistic regression with $L_2$ regularization (detailed list of the operators are given in the supplementary material). For SH-LDM and SH-LDM-RE the predictions are made directly based on the learned Poisson rates of the test set pairs $\{ij\}$, i.e. $\lambda_{ij} = \exp\left(\gamma_i + \gamma_j - \|z_i - z_j\|_2\right)$. Error bars for the following AUC scores were found to be in the scale of $10^{-3}$ and thus provided in the supplementary material.

Results for the moderate-sized networks are given in Table 2. We here observe that the SH-LDM and SH-LDM-RE perform significantly better or on par with the performance of the considered baseline approaches. In particular, the SH-LDM and SH-LDM-RE perform better than all baselines when $D = 2$ which highlights the superiority of LDMs in learning very low-dimensional network representations that accurately accounts for the network structure. We further observe that representing degree heterogeneity with random effects provides extended representational power as the SH-LDM-RE consistently outperforms the SH-LDM. In Figure 2 in the first and second panel we provide the underlying hierarchical structure, as well as an analysis of the accuracy of the SH-LDM when contrasted with the full likelihood evaluated on the moderate-sized network Facebook (results for more networks are provided in the supplementary material). We here observe that the SH-LDM likelihood approximation corresponds well with the true full likelihood providing systematically slightly lower likelihood estimates which we attribute to the SH-LDM inference minimizing the hierarchical approximation. In Figure 2 third panel we observe the predictive performance as a function of latent dimension $D$ and here in general observe that modest improvements in the predictive performance are attained when increasing the embedding dimensions from $D = 2$ to $D = 8$ with no further improvements increasing to $D = 128$ highlighting the efficiency in which SH-LDM and SH-LDM-RE utilize very low-dimensional representations. Results for the large-scale networks are given in Table 3. Again we observe that SH-LDM-RE outperforms the baselines only being surpassed in performance by NETSMF for the *Flixster* dataset. Notably, we again observe very good performance for the SH-LDM-RE, but also NETSMF when utilizing the very low embedding dimension of $D = 2$. In Figure 2, the fourth panel we investigate the convergence of the best performing SH-LDM-RE for the large networks and here observe that the model rapidly converges such that we already after a couple of thousand iterations (scalable regime) obtain competitive performance for link prediction which then gently increases until convergence.

**Node classification:** We assess the success of the proposed framework in the uni-label/multi-label classification task and provide the Micro-$F_1$ scores (Macro-$F_1$ scores are reported in the supplementary). For the experimental setup, we randomly pick $50\%$ of nodes as the training set and use the rest as the testing set. For an accurate comparison across different methods we used two simple classifiers, one linear (logistic regression) and one non-linear (linearithmic k-nearest neighbors classifier (*kNN*)) and reported the highest scores. We found that all methods benefit from using *kNN*. The number of neighbors was set to $k = 10$ (similar results were obtained with higher choices for $k$ as well). Lastly, we report the average Micro-$F_1$ scores across 10 repeated trials. Results

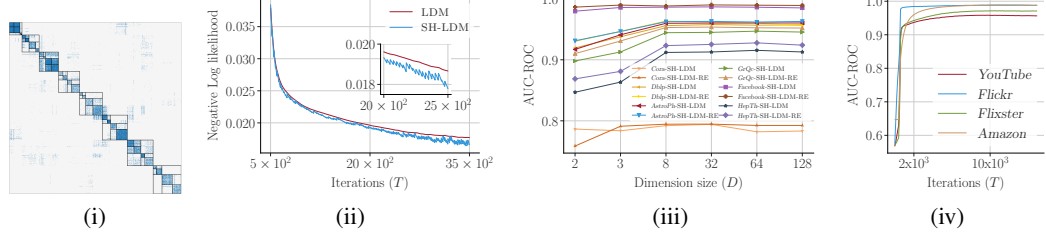

|  |  |  |  |
|---|---|---|---|
| (i) | (ii) | (iii) | (iv) |

Figure 2: (i) *Facebook* ordered adjacency matrix for the learned $D = 2$ embeddings of SH-LDM-RE for a hierarchical structure of $L = 3$. (ii) NLL comparison between SH-LDM and LDM for *Facebook* with $D = 2$. (iii) AUC performance over various networks for varying embedding sizes. (iv) AUC scores of SH-LDM-RE in terms of iterations for large-scale networks.

Table 4: Micro-$F_1$ scores varying embedding dimensions for two moderate and large-scale networks.

|  | *Cora* | | | *DBLP* | | | *Amazon* | | | *YouTube* | | |
|---|---|---|---|---|---|---|---|---|---|---|---|---|
| Dimension ($D$) | 2 | 3 | 128 | 2 | 3 | 128 | 2 | 3 | 8 | 2 | 3 | 8 |
| DEEPWALK | .502 | .712 | **.838** | .519 | .605 | .822 | .231 | .596 | .929 | .293 | .351 | .413 |
| NODE2VEC | .419 | .658 | .835 | .448 | .540 | .815 | .096 | .305 | .895 | - | - | - |
| LINE | .197 | .191 | .794 | .328 | .294 | .771 | .005 | .003 | .003 | .185 | .134 | .177 |
| NETMF | .389 | .653 | .835 | .654 | .707 | .742 | x | x | x | x | x | x |
| NETSMF | .554 | .705 | .842 | .622 | .732 | **.829** | .387 | .649 | .845 | .317 | .361 | .397 |
| RANDNE | .271 | .337 | .731 | .406 | .473 | .718 | .223 | .411 | .787 | .211 | .226 | .277 |
| LOUVAINNE | .804 | .811 | .801 | .780 | .812 | .825 | **.970** | **.971** | **.974** | **.362** | .360 | .359 |
| PRONE | .450 | .611 | .830 | .574 | .634 | .825 | .420 | .750 | .933 | .218 | .274 | .379 |
| SH-LDM | .789 | .807 | .816 | .812 | .814 | .772 | **.970** | **.971** | .931 | .320 | .366 | .414 |
| SH-LDM-RE | **.805** | .809 | .818 | **.805** | **.822** | .808 | .956 | .955 | .931 | .326 | **.367** | **.414** |

for the uni-labeled *Cora* and *DBLP* networks are reported in the two leftmost columns of Table 4. We observe that SH-LDM-RE and SH-LDM significantly outperform the baselines for the regimes of $D = 2, 3$ with only LOUVAINNE being competitive. Results for large-scale and multi-labeled networks *Amazon* and *YouTube* are provided by the two rightmost columns in Table 4. Again, the proposed framework outperforms the baselines for the low-dimensional regime with LOUVAINNE being on par. The superiority of SH-LDM for low-dimensions verifies that homophily and transitivity can express accurately the underlying node class structure.

**Network visualization:** The existing graph representation learning literature mainly focuses on embeddings with dimensionality greater than $D = 2$ and 3. As a direct consequence explanatory analysis and network visualizations rely on dimensionality reduction frameworks, typically using the t-distributed Stochastic Neighbor Embedding (t-SNE) van der Maaten & Hinton (2008). In order to verify the validity of the t-SNE constructed Space (t-SNES), in Figure 3 we provide the labeled-colored True Embedding Space (TES) for $D = 2$, as well as for $D = 2$ and $D = 128$ mapped to $D = 2$ via the use of t-SNE for *Cora* and *DBLP*. We report the performance over the tasks of network reconstruction, clustering and classification respectively reporting the AUC, the Normalized Mutual Information (NMI) and Micro-$F_1$ scores. Our proposed SH-LDM and SH-LDM-RE have the most consistent and reliable scores across both the TES and the t-SNES. For the network reconstruction task we see relatively small decrease in AUC-ROC when transitioning from TES to t-SNES, meaning that the network structure is preserved. For the classification task we observe that SH-LDM SH-LDM-RE benefit from transitioning to t-SNES as the scores for both $D = 2$ and $D = 128$ increase, as summarized in Figure 3 and Table 4. This is not the case for NODE2VEC and LOUVAINNE as their Micro-$F_1$ scores dramatically drop. For most methods NMI benefits when transitioning to t-SNES with SH-LDM giving on par performance for *Cora* while SH-LDM-RE has the highest overall NMI for *DBLP* ($D$=128+t-SNE). We attribute the increase in clustering performance while using high-dimensions and t-SNE (especially for *DBLP*) to the fact that peripheral nodes which act as noise are grouped in a unique cluster, as it is evident from the third and sixth columns in Figure 3. The visualization experiments positions our proposed models as the

silver lining between consistent performance across all GRL downstream tasks as well as network visualizations. All baseline results are available in the supplementary.

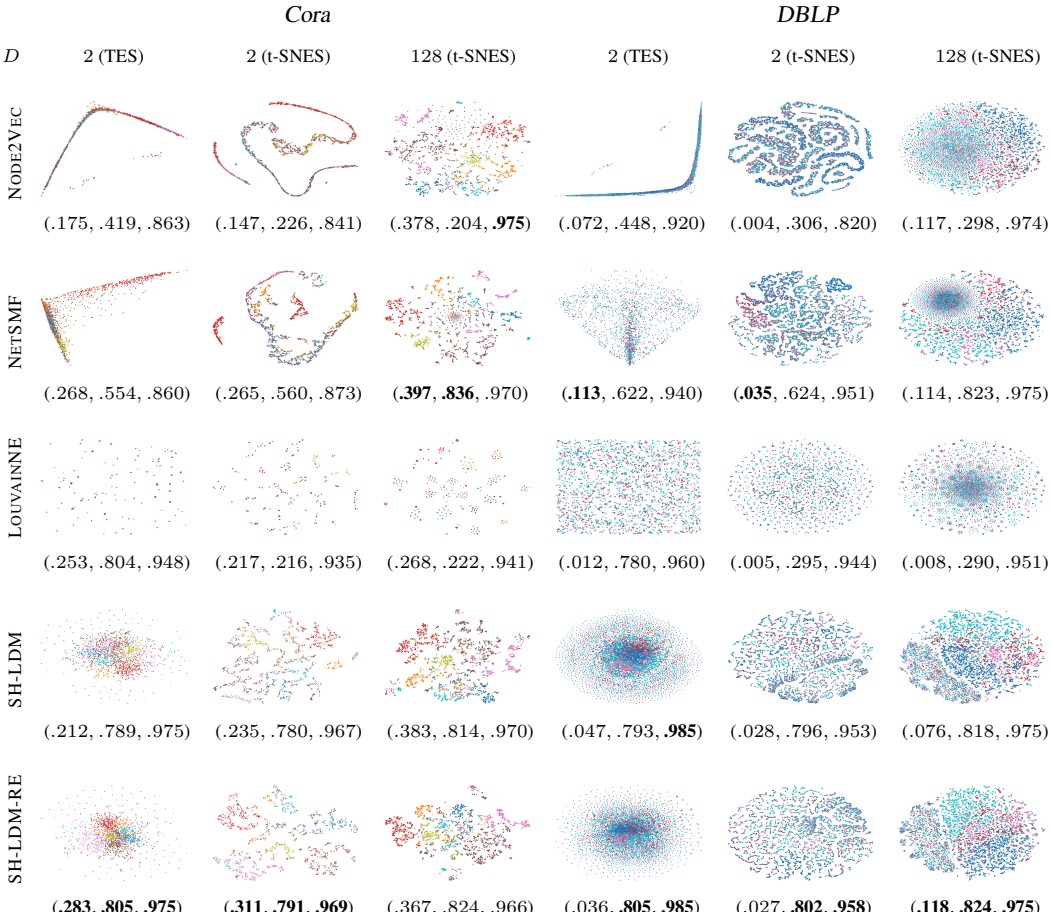

Figure 3: The first two columns for each dataset represent the node embeddings learned in two-dimensional space and t-SNE algorithm is applied for the second and the third columns to reduce the dimension size for the visualization task and to demonstrate the influence of t-SNE algorithm on the embeddings. For each network, NMI (node clustering), Micro-$F_1$ (node classification) and AUC scores (network reconstruction) are reported, respectively.

## 4 CONCLUSION AND LIMITATIONS

We proposed the Scalable Hierarchical Latent Distance Model (SH-LDM), a scalable approach embedding networks using the latent distance model (LDM) enabling characterization of structure at multiple scales. Notably, the approximation provides high accuracy where the likelihood approximation is most important while scaling in complexity by $\mathcal{O}(DN \log N)$. We analyzed ten networks from moderate sizes to large-scale and found that the SH-LDM had favorable performance when compared to existing scalable embedding procedures. In particular, we observed that the SH-LDM well predicts links and node classes utilizing a very low embedding dimension of $D = 2$ providing favorable network visualizations and characterization of structure at multiple scales. The SH-LDM is based on the latent distance model (LDM) and thus good at characterizing transitivity and homophily whereas the random effects enable to take degree heterogeneity into account. Notably, the SH-LDM suffers from the limitations of the LDM and is thus unable to account for stochastic equivalence (Hoff, 2007). Future work should thus investigate how the proposed hierarchical embedding structure can be imposed on more flexible embedding procedures such as the eigenmodel.

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
