# OpenReview forum: "Scalable Hierarchical Embeddings of Complex Networks"
_ICLR.cc/2022/Conference — ICLR 2022 Submitted_

### Official Review · Reviewer_ApfC · 2021-11-01

**Correctness:** 2
**Technical Novelty And Significance:** 2
**Empirical Novelty And Significance:** 2
**Recommendation:** 5
**Confidence:** 4

**Main Review:**

Strengths:
1. It is novel to use LDM for hierarchical node structure representation.
2. Using blocks to approximate O(N^2) and reduce the complexity is reasonable and works well.
3. The proposed method performs well with low embedding dimensions on different datasets.

Weaknesses:
1. It is hard to understand why an auxiliary function is needed for Eq.7 and how it is designed. Motivations and intuitions should be provided.
2. Why the proposed method does not perform well with higher dimensions (e.g., 128D on Cora and DBLP)? More analysis/discussions are expected.
3. Any theory that can explain why SH-LDM performs better with random effects?
4. How does the height L of the hierarchical structure influence the performance of the proposed method?
5. Some important hierarchical embedding approaches are missing as baseline models in the experiments (e.g., HARP and MILE).

Some presentation issues:
1. There is no a, b, c label in Fig. 1.
2. Delete equation label (5).


**Summary Of The Paper:**

The paper studies the node-level representation learning problem. It proposes SH-LDM, which combines the embedding and hierarchical representations for scalable graph representation learning. The hierarchical structure in SH-LDM reduces the time and space complexity of the LDM to linearithmic in terms of the number of nodes. The proposed model works well on link prediction and node classification with low embedding dimensions.

**Summary Of The Review:**

Although the proposed method performs well in experiments under the low embedding dimension setting, more explorations on different settings are expected to understand this model comprehensively. Other concerns are listed above in weaknesses.

---

> ### Author Response · Authors · 2021-11-10
> **Response to ApfC**
>
> We thank ApfC for the constructive feedback and for the assessment of our work.
>
>
> ### 1. It is hard to understand why an auxiliary function is needed for Eq.7 and how it is designed. Motivations and intuitions should be provided.
>
> The classical LDM utilizes a Euclidean distance metric which is not accounted for by normal K-means procedures which utilize a cost function based on the square of the Euclidean norm. In order for our clustering to be consistent with the metric used by LDM, we need to define a clustering procedure that operates on the Euclidean norm, as seen in Equation (7) of the main paper. Such a choice of a cost function for K-means does not lead to analytical expressions or closed-form solution updates. In order to escape from that, we define the auxiliary function in Equation (8) which allows us to define closed-form updates utilizing the perks of having a Squared Euclidean metric. The procedure is described in Section 2.1.2. where we show that minimizing equation J+ wrt. phi we get the correct loss verifying that J+ is indeed an auxiliary function of (7).
>
> ### 2. Why the proposed method does not perform well with higher dimensions (e.g., 128D on Cora and DBLP)? More analysis/discussions are expected.
>
> The proposed model does not need the introduction of such high dimensions since the performance saturates after d=8 dimensions as seen by Figure 2 iii) in the main paper. Including so many residual dimensions can introduce spurious effects on the embeddings modeling noise as signal and lead to minor overfitting. In general, the scope of this paper is to explore low dimensions since for such low dimensions existing GRL methods do not well account for network structure.
>
> We also observe the same trend in many GRL approaches doing the parameter sensitivity analysis for the dimension size. The performance of the models start to decrease after a certain dimension size because of overfitting. Compared to the GRL methods in the literature, we focus on very small dimensions such as 2 and 3 and we demonstrate that we can outperform the classical baseline methods and the recent scalable algorithms.
>
> ### 3. Any theory that can explain why SH-LDM performs better with random effects?
>
> There are existing papers [1,2] introducing random effects to the LDM as referenced in the paragraph before Equation 2 in the main paper. The random-effects account for degree heterogeneity across network nodes.
>
> [1] Peter D Hoff. Bilinear mixed-effects models for dyadic data. Journal of the American Statistical Association, 100(469):286–295, 2005.
>
> [2] Pavel N. Krivitsky, Mark S. Handcock, Adrian E. Raftery, and Peter D. Hoff. Representing degree distributions, clustering, and homophily in social networks with latent cluster random effects models. Social Networks, 31(3):204 – 213, 2009
>
> ### 4. How does the height L of the hierarchical structure influence the performance of the proposed method?
>
> The height of the tree influences linearly the complexity of the clustering procedure.
>
> ### 5. Some important hierarchical embedding approaches are missing as baseline models in the experiments (e.g., HARP and MILE).
>
> The aforementioned algorithms HARP and MILE propose in fact frameworks rather than novel representation learning ideas since they also depend on other GRL methods such as Deepwalk and Node2Vec. Therefore, we preferred to consider the LouvainNE method as a baseline that learns node representations at different levels of the hierarchy and it is also a recent scalable method showing high performance in many tasks. However, we have also started to work on running the MILE and HARP as the reviewer suggested, and hopefully, we will report them soon.

---

> > ### Comment · Reviewer_ApfC · 2021-11-29
> > **Response for authors**
> >
> > Thanks for the response. I am not convinced by some of the responses though, e.g., Q2. The current version needs substantial improvement to meet the ICLR standard.  I will keep the current score.

---

### Official Review · Reviewer_2m7B · 2021-11-02

**Correctness:** 4
**Technical Novelty And Significance:** 2
**Empirical Novelty And Significance:** 2
**Recommendation:** 3
**Confidence:** 4

**Main Review:**

Overall, it seems that the contribution of this paper does not necessarily warrant its publication at CVPR.

Pros:
- The authors develop an unsupervised embedding technique that is scalable.
- Interesting trick: in their model, the probability of the network decomposes over the product of the probability of the individual edges. The main take-away here is that they use a Poisson regression model rather than logistic regression framework. Not sure how this could be applicable to "weighted" graphs, as the authors imply however, but it would have been interesting to delve deeper into the properties of the model as a result of this choice --- it is an interesting choice, that was not sufficiently well motivated.


Cons:
- This is more of a new pipeline rather than a new method --- the authors stitch together existing methods, and while the scalability issue in node embeddings is an interesting aspect, the body of work here is not really novel.
- The method is perhaps not appropriately described --- it should be emphasized that the method for learning structure here is unsupervised (they do not use class labels), which is a big difference with GNNs. Somehow, I thought that the scalability argument was emphasized everywhere, but perhaps not convincing to explain why the method should not be compared with GNNs.
- The main concern with this submission is the method of evaluation:
   1.  The comparison table should list running times and complexities for all of the algorithms, instead of simply the performance.  Currently, we fail to see how "scalable" these truly are, and the performance is only midly impressive.
   2.  The t-sne  evaluation method was not convincing at all, in my opinion. This is (a) a stochastic method, and unless the PCA initialization is used, the results typically vastly vary, and its output is sometimes unpredictable and often sensitive to the choice of perplexity parameter --- I am therefore not quite sure what value to put on the micro-F1 scores, NMI etc.  As a reminder, t-SNE is a visualization method at best, but not a way to do quantitative work. t-SNE is indeed unable to correctly represent local differences in
data density or variance, and cluster sizes and large-scale distances
are not interpretable in t-SNE output embeddings ---- so again, using tsne based distances to assess quality seem inadequate. See [1] and[2] for references on the topic. Instead, statisticians usually prefer using techniques such UMAP rather than t-SNE.


[1] Martin Wattenberg, Fernanda Viégas, and Ian Johnson. How to use t-sne effectively. Distill, 2016.
[2] Nguyen, Lan Huong, and Susan Holmes. "Diffusion t-SNE for multiscale data visualization"


**Summary Of The Paper:**

The focus of the paper is in suggesting a node embedding method that uses hierarchy to ensure scalability. The proposed method, “Scalable Hierarchical Latent Distance Model” ( SH-LDM), aims to reconcile embedding and hierarchical network representations. This method is based on the following components:
-	__Foundational component:__ a Latent Distance Model. Latent Space Models are a special type of node embedding method that uses  the generalized linear model framework to obtain latent node embeddings while preserving network characteristics. In the case of the Latent Distance Models, this means that nodes are placed closer in the latent space if they are closer together. In particular, here, __representing the edges as sampled from a Poisson distribution__ with parameter $\lambda_{ij}$:
$$ \lambda_{ij} = \gamma_i + \gamma_j - || z_i -z_j||_2 $$
Finding a solution in terms of $z_i$ s of this model --- which is not new --- is the object of this paper.

- __Component 2:  Hierarchy:__ To make the fitting of this model more scalable, the authors approximate the distance r $ d(z_, z_j) = || z_i  - z_j|| $ by the distance between centroids $|| \mu_{C_i} -\mu_{C_j}||, \quad \forall C_i\neq C_j$.

- __Component 3: Scalable division of the data in clusters__ The clusters are not known. So the authors use a multiresoluiton KD tree to split the data into K clusters of equal sizes. It further relies on an optimization procedure
for k-means clustering with Euclidean norm utilizing the auxiliary function framework of Tsutsu & Morikawa to perform optimization scalably.
This allows the partition of the data into $K = \log(N)$ clusters.

This results in a reduction of  the total time and space complexity of the LDM to O(N logN)).
The authors then proceed to validate their method using a set of experiments:

 1. node classification

 2. edge prediction

 3. clustering and hierarchical structure recovered in the latent space.

with reasonable performance on a subset of the dataset.


**Summary Of The Review:**

Overall, this paper seems unfit for publication. The novelty is limited, and the methods of validation/ experiments do not make a strong case for the usefulness of such a data processing piepline.

---

> ### Author Response · Authors · 2021-11-10
> **PART1 Response to 2m7B**
>
> We thank 2m7B for assessing the manuscript and for the provided feedback. In the following, we respond to the comments/concerns made by the reviewer.
>
> ### Comment on: Component 3: Scalable division of the data in clusters The clusters are not known. So the authors use a multiresolution KD tree to split the data into K clusters of equal sizes. It further relies on an optimization procedure for k-means clustering with Euclidean norm utilizing the auxiliary function framework of Tsutsu & Morikawa to perform optimization scalably. This allows the partition of the data into K=log(N) clusters.
>
> Our proposed method relies on a partition of K=N/log(N) rather than the K=log(N) falsy stated by the reviewer.
>
> ### Comment on:  Not sure how this could be applicable to "weighted" graphs, as the authors imply however, but it would have been interesting to delve deeper into the properties of the model as a result of this choice --- it is an interesting choice, that was not sufficiently well-motivated.
>
> Since the SH-LDM uses a Poisson likelihood in order to generalize to integer weighted graphs, one simply has to modify the likelihood in Equation 3 expressing with y_ij the weight (or Poisson count) of the node dyad ij. The extra term of the log(y_ij!) present in the likelihood can be disregarded as it is just a constant.
>
>
>
> ### a) This is more of a new pipeline rather than a new method --- the authors stitch together existing methods, and while the scalability issue in node embeddings is an interesting aspect, the body of work here is not really novel.
>
> We would argue that we propose a new method that is not of incremental novelty. It is
> i) the first time the LDM is enhanced with hierarchical representation, as well as the first time a Euclidean k-means clustering procedure is proposed.
> ii) Furthermore, to the best of our knowledge, we propose the first approach demonstrating that the large size networks can be in fact embedded into a very small dimensional space and can provide high accuracy in the downstream tasks such as the link prediction and node classification.
> iii) Contrary to the existing graph representation learning methods in the literature, our approach provides very good visualizations in 2D-space without needing additional post-processing steps such as the t-SNE algorithm.
> iii) Lastly, LDMs have not previously been systematically benchmarked against prominent GRL procedures.
>
>
>
> ### b) The method is perhaps not appropriately described --- it should be emphasized that the method for learning structure here is unsupervised (they do not use class labels), which is a big difference with GNNs. Somehow, I thought that the scalability argument was emphasized everywhere, but perhaps not convincing to explain why the method should not be compared with GNNs.
>
> As the reviewer indicated that most GNN methods are semi-supervised or supervised methods and they can’t show comparable performance to the well-known baseline methods. Their another disadvantage is that they are not applicable for large-scale networks such as Flickr and Flixster. That’s why, we have preferred to adapt the recent scalable methods in addition to the well-known GRL methods. We thank the reviewer and we will update the manuscript by emphasizing the reasons for the baselines.

---

> > ### Comment · Reviewer_2m7B · 2021-11-10
> > **Re: Author's answer**
> >
> > Sorry about the mistake/typo for the number of clusters ---  but that is of little consequence to the review.
> > I am still unsure about the weighted graphs:  it seems to me that most weighted graphs are continuous. Take attention graphs for instance, the weight varies continuously form 0 to 1. How is this compatible with Poisson regression? You'd have to pre-process the data before, but that seems a little unprincipled?
> >
> > For the rebuttal:
> > (a) This does not answer my comment on TSNE. I find this last section perhaps not rigorous enough. What is a good representation? I agree that you can try and quantify that in terms of classification performance (silhouette scores, etc). But (i) some details are missing (how are you clustering), and (ii) i am still not convinced that tsne is the way to go. It is a visualization tool, but I'm not sure that all the metrics you derive from tsne dimensionality reduction, followed by clustering are valid.
> > (b) I would also disagree that GNNs don't scale. Certain methods, such as GraphSage, are notoriously known to be deployable on huge networks.

---

> > > ### Author Response · Authors · 2021-11-11
> > > **Part1: Answer to reviewer's comment on the rebuttal**
> > >
> > > ### Take attention graphs for instance, the weight varies continuously form 0 to 1. How is this compatible with Poisson regression? You'd have to pre-process the data before, but that seems a little unprincipled?
> > >
> > >
> > > It is true that Poisson regression is designed for counting data and thus for integer weighted graphs (a category that covers a plethora of complex networks)
> > >
> > > Since this is essentially a Generalized Linear Model, someone could still model positive continuous weights (via the Poisson GLM) as long as the assumption that the mean is equal to the variance is valid.
> > > It was also shown in [1] that the Poisson target does not necessarily need to be integer-valued for the likelihood estimator to be consistent.
> > >
> > > The negative log-likelihood when disregarding constant terms can also be considered as the objective of non-negative matrix factorization. If we consider the terms $A_{ij}$ and $B_{ij}$ in the KL divergence equation provided in Equation 3 of the work [3] as the dataset value $(y_{ij})$ and $\lambda_{ij}$, respectively, it can be seen that the objective of non-negative matrix factorization corresponds to the negative Poisson log-likelihood even though it is not integer-valued. Therefore, using non-integer values is also valid in terms of the NMF objective.
> > >
> > >
> > >
> > >
> > >
> > >
> > >
> > > ### What is a good representation?
> > >
> > > We would argue that a good representation should:
> > >
> > > a) be Interpretable by human perception, similar nodes are positioned closer in the latent space (one of the main goals and intuition behind GRL).
> > >
> > > b) show good performance in the downstream tasks such as link prediction/network reconstruction/node classification.
> > >
> > > c) not depend on heuristic dimensionality reduction approaches but provide accurate low-dimensional representations with maximum d=3.
> > >
> > > d) provide hierarchical/multiresolutional intrinsic structures of the network facilitating interpretation and visualization at multiple scales.
> > >
> > >
> > > Based on these quality measures, we would strongly argue that SH-LDM returns optimal network representations:
> > >
> > > a) Operates with a Euclidean distance metric naturally modeling human perception on node similarity.
> > >
> > > b) Performs well on all GRL downstream tasks which shows how well it can express the underlying network structure.
> > >
> > > c) Directly and consistently operates on d=2,3 with high performance and thus has no need for heuristic dimensionality reduction approaches.
> > >
> > > d) Naturally represent hierarchical/multiple resolutions based on its block structure and carefully designed/tailored clustering procedure optimized in terms of Euclidean distances to centroids within the latent space.
> > >
> > >
> > > ### i) some details are missing (how are you clustering)
> > >
> > > We perform normal K-means clustering with a number of clusters equal to the network classes.
> > >
> > >
> > > ### (ii) i am still not convinced that tsne is the way to go
> > >
> > > Based on the reviewers’ comments we are trying to verify our results with the proposed UMAP as well. During this experiment, we are not trying to validate our method but criticize and quantify how much researchers can rely on t-SNE for visually assessing their embedding space as it is the main choice in GRL literature. We are arguing that every visualization tool should properly account for network structure.

---

> > > ### Author Response · Authors · 2021-11-11
> > > **Part2: Answer to reviewer's comment on the rebuttal**
> > >
> > > ### (b) I would also disagree that GNNs don't scale. Certain methods, such as GraphSage, are notoriously known to be deployable on huge networks.
> > >
> > > In general, GNN models, particularly, GCN methods provide the scalability by adapting the sampling techniques. For instance, GraphSage, applies neighbood sampling upto r-hop but it leads to the appearance of others problems due to the size of r-hop neighborhood for the networks obeying the small world phenomenon. It can be seen that the time complexity of GraphSage is  O(r^K n d^2) [2] and the time and memory complexity of GraphSage exponentially increases with K and r.
> > >
> > > Although there is a recent trend for the development of scalable GCN methods [4, 5], in general, GCN models highly depend on the node features and they show very poor performance if there is a strong correlation between node features and node labels [6]. On the other hand, our proposed approach outperforms the classical and recent scalable methods without needing the node attributes. By concatenating the embedding variable vectors with the node features together, the proposed approach can also allow the extension of the model with the node features under ​​the metric of Mahalanobis distance (with a block covariance matrix of $[I,0 ; 0,A]$ under the combined node embedding $\tilde{z}_i=[z_i;x_i]$, with $I$ a diagonal matrix, $A$ the covariate coefficients, $z_i$ the latent embeddings and $x_i$ the covariate information for node i.
> > >
> > >
> > >
> > >
> > >
> > >
> > > [1] Gourieroux, C., Monfort, A., & Trognon, A. (1984). Pseudo Maximum Likelihood Methods: Applications to Poisson Models. Econometrica, 52(3), 701–720. https://doi.org/10.2307/1913472
> > >
> > >
> > > [2] Wu, Zonghan & Pan, Shirui & Chen, Fengwen & Long, Guodong & Zhang, Chengqi & Yu, Philip. (2019). A Comprehensive Survey on Graph Neural Networks.
> > >
> > > [3] Daniel Lee, H. Sebastian Seung, (2000), NIPS, Algorithms for Non-negative Matrix Factorization,
> > >
> > >
> > > [4] H. Zeng et al., (2020), ICLR, GraphSAINT: Graph Sampling Based Inductive Learning Method
> > >
> > > [5]  Frasca et al., SIGN: Scalable Inception Graph Neural Networks (2020). ICML workshop on Graph Representation Learning and Beyond
> > >
> > > [6] Duong, Chi Thang et al. “On Node Features for Graph Neural Networks.” ArXiv abs/1911.08795 (2019): n. pag.

---

> ### Author Response · Authors · 2021-11-10
> **PART2 Response to 2m7B**
>
> ### c) The comparison table should list running times and complexities for all of the algorithms, instead of simply the performance. Currently, we fail to see how "scalable" these truly are,
>
> Because of the page limit, in Table 4 in the supplementary, we provide the complexity analysis for all of the considered methods. Our method is expressed with a total complexity of t*N*log(N) where t is the number of iterations of the model. Figure 2 iv) of the main paper shows the effect of iterations with respect to network performance where a small number of iteration defines our method in its ultra-scalable regime while a higher number of iterations boost performance but decreases scalability. We did not include a runtime comparison table in seconds since it is a biased metric for scalability, relying heavily on the programming language/style, parallelization schemes, etc.
>
> ### d) the performance is only midly impressive. + validation/ experiments do not make a strong case for the usefulness of such a data processing pipeline.
>
> It should be noted that our approach targets to learn very small node representations and compared to the baseline methods, it is the best or the second best performing approach across all different network structures and experimental tasks.  The SH-LDM model was found as the only scalable, consistent in performance with ultra-low D=2,3. In addition, the embedding space is highly interpretable due to the Euclidean distance metric, with no need for dimensionality reduction for explanatory analysis.

---

> ### Author Response · Authors · 2021-11-10
> **PART3 Response to 2m7B**
>
> e) The t-sne evaluation method was not convincing at all, in my opinion. This is (a) a stochastic method, and unless the PCA initialization is used, the results typically vastly vary, and its output is sometimes unpredictable and often sensitive to the choice of perplexity parameter --- I am therefore not quite sure what value to put on the micro-F1 scores, NMI etc. As a reminder, t-SNE is a visualization method at best, but not a way to do quantitative work. t-SNE is indeed unable to correctly represent local differences in data density or variance, and cluster sizes and large-scale distances are not interpretable in t-SNE output embeddings.
>
> We strongly agree with the reviewer that t-SNE does not provide a valid method to do quantitative work but it is what GRL literature heavily uses to assess visually the quality of the learned embedding space. As already extensively mentioned and as we emphasized in the manuscript, our proposed method does not require dimensionality reduction visualization frameworks. The aim of the t-SNE study is to quantify the validity of that assessment and to show that in the case of explanatory analysis and visualization (via t-SNE), the constructed t-SNE space of our SH-LDM model is the only one that is consistent with respect to the network structure across GRL downstream tasks. Lastly, we used exactly the same parameter choices for all methods and models, and a normal k-means for clustering based on GRL literature.

---

### Official Review · Reviewer_BVVN · 2021-11-03

**Correctness:** 4
**Technical Novelty And Significance:** 3
**Empirical Novelty And Significance:** 3
**Recommendation:** 3
**Confidence:** 4

**Main Review:**

The main strength of the paper is that it is faster/more scalable than the traditional LSM and that the method appears to perform quite well in experiment.

My main concerns regarding this paper is:

1. The authors introduced several heuristics to approximate things/speed things up: a) approximating the non-link term using blocks b) using a multiresolution KD tree c) minimizing with a Euclidean norm instead of the exact expression, which they propose an auxiliary function optimization procedure for. The main motivation for these heuristics is to reduce run time so that the procedure is scalable. But what the authors did not sufficiently address/rigorously justify is a) how accurate are these approximations? are there any bounds on the approximation error? b) how does the resulting model after applying these changes relate to the original model? are desirable properties that make the LDM useful in the first place retained? Without addressing the above questions, it is very unclear what exactly this new model is capturing and whether it is still an LDM model.

Another way to phrase the above is that the authors proposed modifications to the LDM that are *sufficient* to make it scalable. What the authors did not address is 1) what are the quantifiable tradeoffs that come with this scalability? 2) are these changes to the LDM *necessary* and *unique*? For example, why the multiresolution KD tree? That seemed to have come out of nowhere. Does there exist some other data structure or algorithm that can achieve similar results? If there are alternatives, then the authors should state the alternatives and give reasons for why the choice that they made is the most appropriate given all the alternatives.

A lot of the decisions/choices made in this paper are not sufficiently motivated/justified. Without further justification (especially addressing the uniqueness point above), I find a lot of the decisions (e.g. Using the Euclidean norm as a replacement for the exact expression, and then developing some auxilary function optimization procedure) arbitrary and overengineered.

2. I find it weird that in the experimental section the authors did not compare the SH-LDM with the full LDM. If the SH-LDM is doing so much better than the other competitors, then perhaps the full LDM would be even better, computational costs aside? But I don't recall seeing such results in the literature. If that is the case, then where is the gain of SH-LDM coming from? Is it from the approximations? In any case, I think it is very important for the authors to provide a comparison between their new method and the original method that they modified, even if this means they need to compare them on smaller networks/datasets.

**Summary Of The Paper:**

The authors propose Scalable Hierarchical Latent Distance Model which scales at O(NlogN) rather than O(N^2).

**Summary Of The Review:**

Interesting paper, but lots of decisions/choices that appear arbitrary, insufficiently motivated/justified, and over-engineered.

---

> ### Author Response · Authors · 2021-11-10
> **PART1 Response to BVVN**
>
> We thank BVVN for the evaluation of the paper. We point out each of the reviewers' comments below.
>
> ### 1) The authors introduced several heuristics to approximate things/speed things up
>
> We would strongly argue that our proposed approach is not heuristic, as all modeling choices are carefully designed to assure linearithmic scaling and high accuracy of the approximation. In particular, the method exactly evaluates all links whereas the hierarchical approximation imposed assures high accuracy where most important, i.e. as node pairs are in close proximity. More detailed, Figure 2 i) of the main paper and in Figures 10,11,12 of the supplementary material showing the accuracy of the approximation. Most importantly, our proposed method achieves high performance in a consistent manner, as shown in the experiments section, with no need for heuristic/ad-hoc tuning of the model parameters. This is not the case for many prominent GRL methods such as random-walk based approaches where for example, walk path lengths usually depend on the dataset.
>
>
>
>
> ### a)approximating the non-link term using blocks
>
> The use of block structures to express network structure has been extensively studied and verified in the Stochastic Blockmodel literature. In our paper, the experiments also show how well the network structure can be accounted for using block structures.
>
> ### b)using a multiresolution KD tree
>
>
> We used a similar structure to KD tree for our divisive clustering since such a choice lowers the time and space complexity. The latent space is partitioned based on the Euclidean metric where SH-LDM operates admitting a hierarchical clustering structure accurately accounting for the pairwise Euclidean distances. The choice of data structure is based on the clustering complexity imposed by the model. The conventional k-Means return time complexity of O(t*N*K*D), with t the number of iterations, and space complexity of O(N*K). As extensively analyzed in the main paper (in the last paragraph before subsection 2.1.1), in order for the distance calculation (and thus the model) to return linearithmic complexity we need to introduce K=N/log(N) clusters. The latter enforces clustering complexity of O(N^2) both in time and space, defeating the purpose of this modeling choice. Based on the previous we introduce a KD-tree type of structure since it lowers both time and space complexity of the clustering procedure to linearithmic O(NlogN), making it scalable for large networks.
>
>
> ### c) minimizing with a Euclidean norm instead of the exact expression, which they propose an auxiliary function optimization procedure for
>
>
> We are minimizing the clustering procedure via the minimization of the Euclidean norm since conventional k-means procedures cannot accommodate Euclidean distances used in LDM but only squared Euclidean distances. We therefore carefully design a new Euclidean k-means procedure specifically tailored to the family of Latent Distance Models relying on Euclidean as opposed to square Euclidean distances. The developed procedure derived in Section 2.1.2 achieves Euclidean distance-based clustering with an auxiliary function procedure originally proposed in the context of compressed sensing (citation [56] in the main paper). The proposed clustering procedure thereby provides a carefully designed accurate and scalable approximation to the true distances (all-pairs Euclidean distance matrix used in LDM) with centroids defined directly in the latent space enabling their visualization and interpretation, see also supplementary figures 4,6, and 9.
>
>
> ### d) how accurate are these approximations?
>
> We provide the likelihood comparison between the SH-LDM and the full LDM model in Figure 2 i) of the main paper and in Figures 10,11,12 of the supplementary material showing the accuracy of the approximation.
>
> ### e)  how does the resulting model after applying these changes relate to the original model? are desirable properties that make the LDM useful in the first place retained? Without addressing the above questions, it is very unclear what exactly this new model is capturing and whether it is still an LDM model.
>
> In the supplementary, we perform a comparison with the case-control approach (an unbiased estimate of the full LDM). From Tables 6 to 19 we observe that our SH-LDM has on par performance, thus verifying that all the LDM properties are retained. This is also evident from the ordered adjacency matrices in Figure 2. i) of the main paper and Figures 1,7,8 in the supplementary where we clearly see the homophilic structure obtained while link prediction performance verifies transitivity.

---

> ### Author Response · Authors · 2021-11-10
> **PART2 Response to BVVN**
>
> ### 2) Another way to phrase the above is that the authors proposed modifications to the LDM that are sufficient to make it scalable. What the authors did not address is 1) what are the quantifiable tradeoffs that come with this scalability? 2) are these changes to the LDM necessary and unique? For example, why the multiresolution KD tree? That seemed to have come out of nowhere. Does there exist some other data structure or algorithm that can achieve similar results? If there are alternatives, then the authors should state the alternatives and give reasons for why the choice that they made is the most appropriate given all the alternatives.
>
> We adopt a multiresolution-KD tree type of data structure due to the divisive partition it enforces in the latent space. We discuss in Section 2.1.2 how agglomerative hierarchical clustering has a prohibitive complexity and we, therefore, rely on a divisive k-means type of procedure. Using a tree structure for our divisive clustering also provides approximations of increasing accuracies. That as we move down the tree and clusters contain fewer and fewer nodes the centroid describes better the all-pairs distance between clusters when nodes are in close proximity and the accuracy of the approximation is the most important. Based on the homophilic nature of the SH-LDM similar nodes are thus positioned in finer clusters providing better approximations to distances of nodes that are strongly related and that highly affect the total likelihood.
>
> ### 3)I find it weird that in the experimental section the authors did not compare the SH-LDM with the full LDM. If the SH-LDM is doing so much better than the other competitors, then perhaps the full LDM would be even better, computational costs aside? But I don't recall seeing such results in the literature. If that is the case, then where is the gain of SH-LDM coming from? Is it from the approximations? In any case, I think it is very important for the authors to provide a comparison between their new method and the original method that they modified, even if this means they need to compare them on smaller networks/datasets.
>
> The LDM has not been well established in the GRL literature (e.g. never been used for node classification) and has not been benchmarked across prominent GRL methods nor for large-scale networks. Benchmarking LDM is another contribution of the current study. In addition, it is the first time that LDM can achieve high accuracy in link prediction and node classification for large networks using very low dimensional representations enabling accurate large-scale network visualization in two or three dimensions without relying on heuristic post-processing procedures such as t-SNE. Based on our comparison with the case-control approach we see that SH-LDM performs on par with a full LDM as provided in the supplementary material.

---

### Official Review · Reviewer_MkFe · 2021-11-03

**Correctness:** 3
**Technical Novelty And Significance:** 2
**Empirical Novelty And Significance:** 2
**Recommendation:** 3
**Confidence:** 3

**Main Review:**

The LDM approach is a relatively old approach to learning latent representations for networks. The innovation of this paper is instead of evaluating the exact likelihood function that involves computing pairwise distances among nodes to resort into a clustering technique: Group nodes based on their respective embeddings and only compute the pairwise distances between the centroids of each cluster. This allows for the term that contributes to O(N^2) in the evaluation to reduce to O(K^2).
In the rather rough analysis of the runtime the authors make the following assumptions:
i) The network is sparse -- this is a valid assumption however it should be made clear that the claimed running time of O(nlong) is made based on this assumption
ii) The embedding dimensions is a constant -- which  follows clearly by setting it to a really small value, like 2.
iii) The cluster sizes are equal.
I have a two-fold objection with the last assumption:
1. If we think of networks as collections of clusters, then it is usually the case that these are highly imbalanced. E.g., in several empirical studies as in [1], we see that usually there is a collection of small large clusters and an abundance of smaller clusters (called whiskers) that are connected through the core clusters. I am surprised therefore that this approach can lead to good performance.
2. Another objection is that the authors do not make clear how they get this clustering. Initially, the use of a multiresolutional KD-tree is proposed. However, if we stop splitting nodes, when we reach K nodes in the tree, then there is no guarantee that the resulting clusters will be of equal size. The splitting criterion can be arbitrary (is not defined here), while a good empirical practice is to split along the dimension of higher variance -- a balanced partitioning is guaranteed only for the first split, then it depends. As this approach also fails in terms of reducing the computational burden, the authors try to propose other approaches, like a modified k-means or a balanced binary. The latter of course could work, but there is no information at all on how this is created: How de we choose which points (nodes) to fall under the  left or right child of a split. The paragraph above 2.2 that tries to describe this procedure is rather confusing with limited information. As a reference, a method that tries to achieve balanced partitioning of nodes is described in [2].

Also, while the evaluation of the  likelihood function is discussed in detail, the same is not true for the convergence of the model. How many iterations do we need and how much time does it take to train the model?

In other minor remarks:
1. Defining in the abstract that N is the number of nodes could help exposition for those not familiar with network terminology.
2. In page 3, $\theta$ is not defined.
3. As part of the LL function is evaluated between cluster centroids, is the model having some relationship with the SBM models?
4. Figure 1 does not have indexing.
5. The same Eq. is labeled as (5) & (6).
[1] Leskovec, J., Lang, K. J., Dasgupta, A., & Mahoney, M. W. (2009). Community structure in large networks: Natural cluster sizes and the absence of large well-defined clusters. Internet Mathematics, 6(1), 29-123.

[2] Karypis, G., & Kumar, V. (1997). METIS: A software package for partitioning unstructured graphs, partitioning meshes, and computing fill-reducing orderings of sparse matrices.

**Summary Of The Paper:**

The authors of this paper propose using Latent Distance Modeling (LDM) for embedding networks. LDM is an old model and simply relies on estimating the probability of an edge based on the distance of the respective embeddings of the two endpoints (+ some fixed-effect terms that capture the degree nonhomogeneity. The problem with this approach is that evaluating the likelihood function to maximize requires computing all pairwise distances between nodes in the graph -- thus scales quadratically with the size of the graph.
To tackle this problem the authors propose grouping nodes (based on their embeddings in each iteration) into clusters and only evaluating the pairwise distance between the centroids of the clusters. This allows for faster evaluation of the likelihood function. Also, according to the authors, the LDM model (with and without fixed effects terms for each node), is able to achieve higher performance in different tasks (classification and link prediction) than other popular embedding methods (like DeepWalk or Node2Vec) using just 2 to 3 dimensions for the node embeddings. As a reference, the SoA approaches usually use 128 dimensions to achieve good performance.

**Summary Of The Review:**

The paper proposes a centroid-based approximation to evaluating the likelihood function of latent models -- a rather incremental approach. The main problem is that the final choice of the clustering procedure -- which is the crux of this paper -- is not clearly described. The authors describe more in detail other appoaches -- like KD-trees -- that fail, whereas for the actual choice -- balanced binary tree?-- we know little to nothing. Moreover, the clustering approach itself, and the need for the clusters to be balanced, is one that I have objections based on reasons described before. Under the assumptions that we use O(long) balanced clusters, then (n/logn) nodes will fall under the same cluster. Hence, almost all nodes! Maybe then the relevant term that gives this O(N^2) burden is not relevant to optimize? For these reasons, and despite the claimed superior performance in downstream tasks, I am not in favor of recommending this paper for acceptance in its current form.

---

> ### Author Response · Authors · 2021-11-10
> **PART1 Response to MkFe**
>
> We thank MkFe for the assessment of the manuscript and for feedback. Below we address the reviewer's comments point by point.
> ### i) The network is sparse -- this is a valid assumption however it should be made clear that the claimed running time of O(nlogn) is made based on this assumption
>
> We will emphasize more the assumption in the paper. Indeed our proposed approach defines a linearithmic complexity for the likelihood calculation when the network is sparse. This is also highlighted in section A.8 of the supplementary and more specifically Figure 3 shows that the proposed approach scales linear with the number of edges in the network.
>
> ### ii) The embedding dimensions is a constant.
>
> This assumption is valid and highlights the advantages of using LDMs where homophily and transitivity can be expressed with very low dimensions and thus disregarded from the complexity analysis. The low-dimensional node representations are also one of the strongest points of the manuscript which surprisingly shows that very large networks can in fact be embedded into very small dimensional spaces and can lead to high performance in downstream machine learning tasks. This is a very significant observation that we have not seen before in the literature to the best of our knowledge.
>
>
> ### iii) The cluster sizes are equal.
>
> This is not the case, the only constraint we impose on the clusters is that if they reach the size of log(N) points they are considered as leafs and thus are not split in the next layer of the tree. We think the misunderstanding stems from the example given by Figure 1 where we show equally sized clusters. This choice was only made in order to simplify the visualizations. The resulting hierarchies were provided in Figure 2. i) of the main paper, as well as, Figures 1,6,7,8,9 in the supplementary material highlight that the partitions are not equally sized. In principle, we could enforce equally-sized clusters across the divisive partition, such a choice though would affect homophily and transitivity by occasionally positioning nodes that are not similar in the same cluster. As an immediate result of the uneven clusters, the (after the first layer) binary tree will not be balanced but we see empirically that the height of the tree is equal to c*log(N) with c<<N ensuring linearithmic complexity and thus making our analysis in section 2.1.2 accurate. The expressive power of the clustering procedure and how it manages to model homophily and transitivity can be seen empirically from the off-diagonal parts of the ordered adjacency matrices in Figure 2. i) of the main paper and Figures 1,7 and, 8 in the supplementary material.

---

> ### Author Response · Authors · 2021-11-10
> **PART2 Response to MkFe**
>
> ### 1)If we think of networks as collections of clusters, then it is usually the case that these are highly imbalanced. E.g., in several empirical studies as in [1], we see that usually there is a collection of small large clusters and an abundance of smaller clusters (called whiskers) that are connected through the core clusters. I am surprised therefore that this approach can lead to good performance.
>
>
> Our approach relies on a divisive clustering over the latent space and the number of clusters does not represent the number of network communities. The initial space is split into log(N) coarse clusters and then each of these coarse clusters is continuously binary split until each of the remaining clusters contains a maximum of log(N) points. The scenario of highly imbalanced clusters only affects the height of the tree. Therefore, the claim that the abundance of small clusters can affect performance is erroneous.
>
>
>
> ### 2)Another objection is that the authors do not make clear how they get this clustering.
>
>
> The splitting criterion is defined in the same nature as the conventional divisive K-means procedure. The framework is analytically described in Section 2.1.2. In each binary split, the centroids are initialized based on the parent centroid plus some small noise which are then updated as in the conventional K-means algorithm. Essentially, every binary split is a clustering procedure over the parent node population into two new centroids, giving an iterative clustering (divisive) procedure with the cost function of Eq. 8 in the main paper. Unfortunately, conventional k-means procedures cannot accommodate for Euclidean distances used in LDM but only squared Euclidean distances. We therefore carefully design a new Euclidean k-means procedure specifically tailored to the family of Latent Distance Models relying on Euclidean as opposed to square Euclidean distances. The developed procedure derived in Section 2.1.2 achieves Euclidean distance-based clustering utilizing an auxiliary function procedure originally proposed in the context of compressed sensing (citation [56] in the main paper). The proposed clustering procedure thereby provides a carefully designed accurate and scalable approximation to the true distances (all-pairs Euclidean distance matrix used in LDM).
>
> ### 3)" The main problem is that the final choice of the clustering procedure -- which is the crux of this paper -- is not clearly described. The authors describe more in detail other appoaches -- like KD-trees -- that fail, whereas for the actual choice -- balanced binary tree?-- we know little to nothing"
>
> In our paper we still adopt a KD-tree type structure where partitions on the latent space are based one the Euclidean distance between nodes. We argue how agglomerative hierarchical clustering has a prohibitive complexity and we therefore rely on a divisive k-means type of procedure, please see section 2.1.2. In the final version of the paper, we will clarify these points in order to alleviate the confusion.
>
> ### 4)"Moreover, the clustering approach itself, and the need for the clusters to be balanced, is one that I have objections based on reasons described before. Under the assumptions that we use O(long) balanced clusters, then (n/logn) nodes will fall under the same cluster Hence, almost all nodes!"
>
> The reviewer missed that the number of final node clusters is set as K=N/log(N) with every leaf cluster having assigned log(N) nodes to ensure linearithmic complexity. Revisiting again Figure 1 on the main paper we see how in the coarse level we split initially to log(64)=4 clusters and each of these clusters are sequentially binary split until we reach the leaf clusters (red cells) containing log(64)=4 nodes. In a real dataset with uneven clusters, the total number of clusters K is approximately N/log(N) but the number of nodes in leaf clusters will deterministically have log(N) as an upper bound.

---

> > ### Comment · Reviewer_MkFe · 2021-11-20
> > **response to authors**
> >
> > Thanks for clarifying certain parts of your clustering approach.
> > Regarding the issue with the number of clusters + their size: I am not implying that they necessarily should follow the underlying communities. However, as a latent space model, there should be some correlation. For example, consider a clique -- all nodes could possibly have roughly the same representation (+some noise with regards to which nodes they connect). This is I guess, the motivation of this work -- however, if there is no consistency, between the underlying structure and the way the clusters are formed, I can not see how we can have both good link prediction performance + classification.
> > Another issue that remains unanswered is an actual running time comparison between the approaches.

---

> > > ### Author Response · Authors · 2021-11-20
> > > **Response to reviewer**
> > >
> > > We thank MkFe for answering our rebuttal.
> > >
> > > ###  if there is no consistency, between the underlying structure and the way the clusters are formed, I can not see how we can have both good link prediction performance + classification
> > >
> > > The proposed method provides a block-structure calculation only for the second (non-link term) of the likelihood in Eq. 3 of the main paper while the first term (link term) of the likelihood is calculated analytically as in the Latent Distance Model proposed by Hoff et al. This characteristic guarantees the properties of homophily, transitivity and for the structure inside the clusters to be preserved. In addition, our experiments on 10 different networks of various sizes and structures verify the representation power of the proposed model.
> > >
> > > We would like to highlight the use of a kNN classifier which is a non-linear classifier respecting local structures. In addition, we observed consistent and high performance across 4 networks of different sizes, confirming the ability of the proposed model to cluster nodes with just two latent dimensions.
> > >
> > >
> > > ### Another issue that remains unanswered is an actual running time comparison between the approaches.
> > >
> > > Because of the page limit, we have provided the complexity analysis for all of the considered methods in Table 4 of the supplementary materials. Our method is expressed with a total complexity of TNlog(N) where T is the number of iterations of the model. Figure 2 iv) of the main paper shows the effect of iterations with respect to network performance where a small number of iterations defines our method in its ultra-scalable regime while a higher number of iterations boost performance but decrease scalability. We did not include a runtime comparison table in seconds since it is a biased metric for scalability, relying heavily on the programming language/style, parallelization schemes, the environment etc.
> > >
> > > Another advantage of our method compared to the baseline method is that we demonstrate that real-world networks can be in fact embedded into 2 or 3 dimensional spaces with high performance in the downstream tasks. Therefore, the dimensionality parameter of our approach can be also discarded in the complexity analysis.

---

### Decision · Program_Chairs · 2022-01-20

**Decision:**

Reject

**Comment:**

This paper proposes SH-LDM, which approximates the LDM model with a hierarchy of clusters. The authors should discuss the details about clustering and how this algorithm can benefit from sparsity in a more rigorous language.

The authors should review the rich literature on scaling up distance-based methods such as kNN and kernel methods, which this paper belongs to. The title is also misleading; the paper mainly discusses scalable link prediction rather than learning new embeddings.

The reviewers have raised several questions about the experiments. For example, the main results should be a table for comparing the speed rather than the accuracy of the algorithms. Also, the original LDM should be included in the accuracy tables. The settings in the experiments, such as embedding dimensions, are not appropriate for large graphs.